# Examining the association of clinician characteristics with perceived changes in cervical cancer screening and colposcopy practice during the COVID-19 pandemic: a mixed methods assessment

**Lindsay Fuzzell[1]\*, Naomi C Brownstein[2], Holly B Fontenot[3], Paige W Lake[1], Alexandra Michel[3], Ashley Whitmer[1], Sarah L Rossi[4], McKenzie McIntyre[1], Susan T Vadaparampil[1,5], Rebecca B Perkins[4]\***

[1]H. Lee Moffitt Cancer Center & Research Institute, Health Outcomes and Behavior, Tampa, United States; [2]Medical University of South Carolina, Charleston, United States; [3]University of Hawaii at Manoa, Nancy Atmospera-Walch School of Nursing, Honolulu, United States; [4]Boston University, Chobanian & Avedisian School of Medicine, Boston, United States; [5]H. Lee Moffitt Cancer Center & Research Institute, Office of Community Outreach, Engagement, and Equity, Tampa, United States

**\*For correspondence:**
Lindsay.Fuzzell@moffitt.org (LF);
rbperkin@bu.edu (RBP)

**Competing interest:** The authors declare that no competing interests exist.

## Abstract

**Background:** The COVID-19 pandemic led to reductions in cervical cancer screening and colposcopy. Therefore, in this mixed method study we explored perceived pandemic-related practice changes to cervical cancer screenings and colposcopies.

**Methods:** In 2021, a national sample of 1251 clinicians completed surveys, including 675 clinicians who performed colposcopy; a subset (n=55) of clinicians completed qualitative interviews.

**Results:** Nearly half of all clinicians reported they were currently performing fewer cervical cancer screenings (47%) and colposcopies (44% of those who perform the procedure) than before the pandemic. About one-fifth (18.6%) of colposcopists reported performing fewer LEEPs than prior to the pandemic. Binomial regression analyses indicated that older, as well as internal medicine and family medicine clinicians (compared to OB-GYNs), and those practicing in community health centers (compared to private practice) had higher odds of reporting reduced screening. Among colposcopists, internal medicine physicians and those practicing in community health centers had higher odds of reporting reduced colposcopies. Qualitative interviews highlighted pandemic-related care disruptions and lack of tracking systems to identify overdue screenings.

**Conclusions:** Reductions in cervical cancer screening and colposcopy among nearly half of clinicians more than 1 year into the pandemic raise concerns that inadequate screening and follow-up will lead to future increases in preventable cancers.

**Funding:** This study was funded by the American Cancer Society, who had no role in the study's design, conduct, or reporting.

## Editor's evaluation

This important work provides evidence regarding the impact of the COVID-19 pandemic on cervical cancer screening and precancer treatments in the USA. As there are few screening registries, the study provides solid evidence using a survey of health providers' impressions to assess whether

cervical cancer screening services declined during the pandemic. The work will be of interest to public health professionals working in cancer prevention.

## Introduction

Cervical cancer prevention programs have been among the most successful cancer prevention programs to date (*Sawaya and Huchko, 2017*). In the past decade, the addition of routine human papillomavirus (HPV) testing to screening programs has allowed safe extension of screening intervals through greater reassurance against subsequent cancer development among patients with negative results, and also led to more precise management of patients with abnormal results (*Schiffman et al., 2011*; *Schiffman et al., 2018*; *Castle et al., 2018*). However, longer screening intervals may lead to underscreening if patients are not recalled on schedule, and patients with high-risk medical conditions or prior abnormal screening histories need more frequent testing. The coronavirus 2019 (COVID-19) pandemic impacted the ability to perform routine cancer screenings, which may threaten progress made to date at reducing cervical cancer incidence and mortality (*Wentzensen et al., 2021*).

At the onset of the pandemic, cancer screenings decreased substantially (*Chen et al., 2021*; *Poljak et al., 2021*; *Amram et al., 2022*; *Smith and Perkins, 2022*). Nationwide, cervical cancer screening rates fell rapidly in 2020 compared with previous years (*Miller et al., 2021*; *Mayo et al., 2021*). Limited evidence also suggests that colposcopy procedures were impacted during this time, though US data are lacking (*Istrate-Ofițeru et al., 2021*; *Masson, 2021*). As the pandemic has progressed, cancer screening rates have begun to rebound (*Chen et al., 2021*; *McBain et al., 2021*), but considerable challenges are still present. Initially patient fear and closed clinics affected ability to perform cervical cancer screening and colposcopy (*Massad, 2022*). Currently, lower screening rates continue due to high turnover and medical staff shortages, as well as longer wait times for scheduling appointments due to backlogs (*Wentzensen et al., 2021*; *Smith and Perkins, 2022*; *Massad, 2022*). Few studies have explored the impact of the COVID-19 pandemic on clinician perceptions of cervical cancer screening (*Price et al., 2022*) and colposcopy rates compared with prior to the pandemic. This paper examines the quantitative association of clinician characteristics with perceived changes in screening and colposcopy during the pandemic period. Additionally, through qualitative interviews, we explored how clinicians experienced pandemic-related changes in screening and colposcopy.

## Methods

### Participant recruitment

Participant recruitment is detailed elsewhere (*Vadaparampil et al., 2023*). Briefly, clinicians were eligible to participate if they were: (1) a physician or advanced practice provider (APP) (nurse practitioner [NP], physician assistant, or certified nurse midwife) practicing in internal medicine, family medicine, obstetrics and gynecology (OB/GYN), or women's health; and (2) performed cervical cancer screening. Data were collected between March-August 2021 (surveys) and June-December 2021 (interviews). For context, the COVID-19 vaccine became available to healthcare providers in the US in early 2021. The US general public had widespread access to vaccination beginning in the summer of 2021. By the fall of 2021, the pandemic appeared to be less acute in the US, with healthcare organizations attempting to resume normal operations through the end of the year. Masking, social distancing and reduced capacities indoors, and enhanced cleaning procedures were public health practices in place with varying levels of intensity across the US at this time. Between March and August 2021, we recruited clinicians from: the National Association of Nurse Practitioners in Women's Health (NPWH) email listserv, a healthcare physician panel representing a variety of specialties via Dynata (an online market research firm), and the American Society for Colposcopy and Cervical Pathology (ASCCP) mailing list. NPs were recruited via email blasts to NPWH listserv members (~N = 2500; ~20% response rate). ASCCP members were recruited via an external mail house using a protocol based on Dillman's Total Design Method (N=1000; 21.8% response rate) (*Dillman, 1978*). An additional ~250 OB/GYNs and ~250 Internal Medicine and Family Medicine physicians were recruited using Dynata (response rate not available). All participants were compensated. Study participants from all three sources who completed the quantitative survey were asked if they would be willing to participate in phase two of

the study that included a qualitative interview. A random sample of those who indicated willingness were later contacted for participation.

## Survey content and study variables

As previously described (*Vadaparampil et al., 2023*), survey questions were based on Cabana's Guideline Based Practice Improvement Framework (*Cabana et al., 1999*) and previous research by study co-investigators (*Perkins et al., 2020*; *Malo et al., 2016*). An expert panel (n=10), including physicians and APPs from multiple specialties reviewed the survey, and the survey was refined based on their feedback. Finally, the survey was piloted with target clinicians (N=27), revised, finalized, and distributed between March and August 2021. The survey covered several areas related to cervical cancer screening practices and management of abnormal screening results, including presentation of vignettes focused on screening intervals, management or treatment, and screening exit or continuation in relation to 2019 ASCCP risk-based management guidelines adoption, as well as a subset of items for clinicians who perform colposcopy. There were also items related to HPV self-sampling, as well as the impact of the COVID-19 pandemic on screening and follow-up (which is the focus of the present manuscript).

### Clinician and practice characteristics

Age was measured in years and grouped into four categories. Gender identity was assessed as male, female, transgender, and other. Race was categorized as (1) Asian, (2) Black/African American, (3) White, and (4) mixed race, Native Hawaiian/Pacific Islander, American Indian/Alaska Native, other. Ethnicity was identified as Hispanic/Latinx or non-Hispanic/Latinx. For all variables that allowed write-in/free responses, we individually examined responses to determine if they could be accurately re-classified within the pre-determined categories for each variable.

Medical training was assessed as physician (MD, DO) or APP. Medical specialties were OB/GYN, family medicine, and internal medicine for physicians, and women's health for APPs only. We combined training and specialty variables to create one clinician type variable with four groups: OB/GYN physicians, family medicine physicians, internal medicine physicians, and APPs. Practice type included: (1) academic medical center, (2) hospital-based practice (including hospitals and military, post-operative care, and long-term care facilities), (3) private practice/group practice, (4) community health/safety net setting (included federally qualified or community health centers, planned parenthoods, public health departments, and college health centers). Geographic location included four US regions (Northeast, South, Midwest, West); 9% of respondents who did not provide state or zip code were classified as non-responders.

### COVID-19 and pandemic-related behaviors and practice patterns

The survey item used for our primary outcome assessed perceptions of how the pandemic affected cervical cancer screening practices (doing fewer; the same number; or more HPV screens than before the pandemic). Participants were also asked to indicate whether they performed colposcopy (yes/no). Those who performed colposcopy then answered questions on how the pandemic affected their practices for (1) colposcopy (doing fewer; the same number; or more colposcopies than before the pandemic); and (2) loop electrosurgical excision procedure (LEEP) (provided LEEP on site before the pandemic and still doing so at same capacity; provided LEEP on site before the pandemic and still doing but at reduced capacity; provided LEEP on site before the pandemic and now are referring to another facility; have always referred to another facility for LEEP and continue to do so).

## Qualitative interview development, content, and interview processes

The qualitative interview guide was developed based on Cabana's Guideline Based Practice Improvement Framework (*Cabana et al., 1999*). The draft interview guide was reviewed by an expert panel (n=7) including clinicians from multiple primary care specialties. The interview guide was then refined based on expert feedback, pilot tested in a mock interview, further revised, and finalized. The final interview guide included in-depth exploration of cervical cancer screening and management items explored in the quantitative survey. We more deeply explored screening practices (barriers and facilitators to screening for each clinician's patient panel), adherence to 2019 ASCCP guidelines (how clinicians assess if patients are due for screening, type of screening

test used, screening interval used and reasoning) barriers to adoption of ASCCP guidelines, HPV self-sampling (benefits and concerns), and the impact of the pandemic on screening and management practices. Additionally, there was a subset of questions for colposcopists (on colposcopy training, LEEP self-performance versus referral, biopsy location). This manuscript focuses on qualitative findings relevant to the COVID-19 pandemic and its impact on screening and abnormal results follow-up (pause and resumption of screening or follow-ups during pandemic, catching up on missed screenings). Pandemic-related items focused on how the pandemic changed cervical cancer screening practices, pauses to screening or abnormal follow-up (colposcopy or treatment services) approaches for patients who missed screening or follow-up appointments during the pandemic, including strategies for re-engagement. Three co-authors (HF, RBP, AM) trained in qualitative methodology and with expertise in cervical cancer screening conducted qualitative interviews via video conference between June and December 2021. Interviews were audio recorded and transcribed verbatim.

This study was approved by Moffitt Cancer Center's Scientific Review Committee and was reviewed by an Institutional Review Board. The study was given exempt determination by Moffitt's IRB, Advarra (MCC #20048), and Boston University's IRB (BMC IRB# H-41533). All study participants viewed (for surveys) or were read (for interviews) an information sheet in lieu of reading and signing an informed consent form.

## Analytic plan

### Quantitative analyses

We assessed descriptive statistics of clinician and practice characteristics and behaviors. We conducted separate binomial logistic regressions examining the associations of clinician and practice characteristics with responses to items assessing the impact of the pandemic on reported number of cervical cancer screening and on colposcopies (doing the same or more versus fewer than before the pandemic). Age, race, ethnicity, gender, region, clinician type, and practice type were included in the full model for each outcome, as clinician characteristics have previously been associated with cervical cancer screening practices (*Almeida et al., 2013*; *Becerra-Culqui et al., 2018*; *Haas et al., 2021*). For all logistic regression models, we used manual backward selection to individually remove variables exceeding a p-value of 0.10 from each model, but determined a priori that clinician type, practice type, and region would be retained in all models regardless of the corresponding p-values based on the importance of these factors in determining screening and colposcopy practices during the pandemic (*Almeida et al., 2013*; *Becerra-Culqui et al., 2018*; *Haas et al., 2021*; *Horner et al., 2011*). Given the few studies that have explored factors associated with clinician perspectives of changes in cervical cancer screenings and colposcopies during the pandemic, we selected a value for inclusion and significance of 0.10. This strikes a balance between the commonly accepted method of using the AIC (Akaike's information criterion, which implicitly assumes a significance level of 0.157), and the often-used significance level of 0.05. Quantitative analyses were conducted in SPSS Version 26.

### Qualitative analyses

Pandemic-related qualitative interview items were coded using thematic content analysis (*Elo and Kyngäs, 2008*). A priori codes were developed based on the questions in the initial interview guide and a codebook was developed to operationalize and define each code. The qualitative analysis team independently reviewed the data twice. In the first coding pass, the team hand-coded the data with the initial codes and made notes on possible new codes. After the first round of coding, they discussed notes on possible new codes. After reaching consensus, the codes were revised and they again independently reviewed the transcripts and updated code categories from the first coding pass. The second coding pass serves to 'clean up' codes unanticipated in the first coding pass and identify emergent themes not identified in the initial coding scheme (*Krueger, 1998*). All transcripts were coded by at least two coders. Coding discrepancies were resolved by discussion in weekly group meetings to achieve consensus. Coding was conducted in a shared data sheet for ease of completing coding in a centralized database across varying institutions.

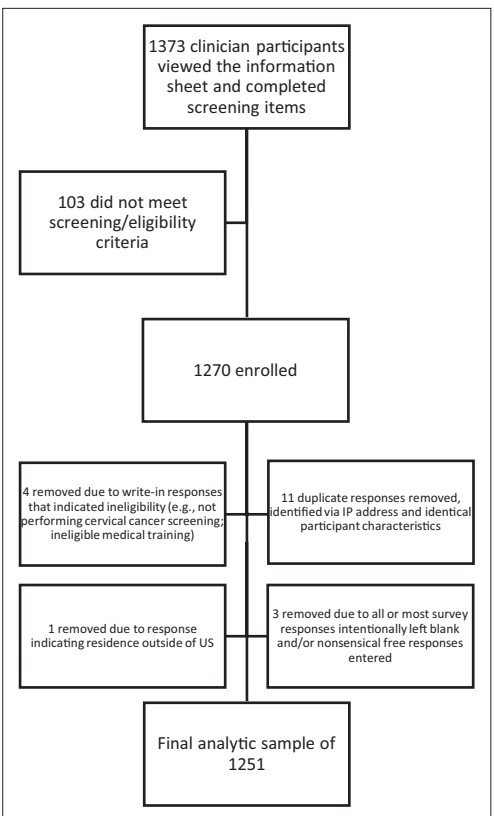

**Figure 1.** Flow diagram depicting potentially eligible, enrolled, and final analytic samples.

## Results

### Quantitative findings

Potential participants (N=1373) viewed the study information sheet and completed screening items; 103 were ineligible due to not performing cervical cancer screening or having an ineligible training/specialty (e.g., pharmacy). Nineteen additional responses were removed as duplicates, nonsensical write-in responses, or ineligibility not previously identified via demographic items, resulting in a final sample of 1251 participants (509 recruited via NPWH [web], 524 from Dynata [web], and 218 from ASCCP [204 mail, 14 web]). See *Figure 1*.

*Table 1* describes clinician practice characteristics, behaviors, and screening practices. The total clinician sample was primarily White (77.8%), non-Hispanic (91.9%), and female (74.7%), with adequate representation from each age group. Regions of practice were distributed across the US (Northeast 18.4%, South 28.9%, Midwest 21.7%, West 22.1%, no response 9%). About half of participants were women's health NPs or other APPs specializing in women's health (48.7%), one-quarter were OB/GYN physicians (26.6%), and the remainder were internal (8.7%) and family medicine (16.0%) physicians. Over half of clinicians (54.0%, n=675) indicated that they perform colposcopies. Colposcopist characteristics were generally comparable to those of the full sample (*Table 1*). Of note, colposcopists were slightly older (32% aged 50–59; 29% aged 60+), and more often OB/GYN physicians (47.0%), compared with the full sample.

*Table 2* details participants' perceptions of their performance of screening, colposcopy, and LEEP at the time of the survey, which was conducted between March and August 2021 (fewer, same, more than before the pandemic). Responses indicated that, over 1 year into the pandemic, 47% of all clinicians reported they were currently performing fewer cervical cancer screenings than before the pandemic and 44.1% of colposcopists were performing fewer colposcopies than prior to the pandemic. Among colposcopists, about one-fifth reported disruptions in LEEP; 18.6% reported performing fewer LEEPs than prior to the pandemic, while 1.3% reported no longer being able to offer LEEP at their facility and therefore referring all patients out who required this service. The remaining colposcopists either reported performing LEEP at the same level (51.1%) or continued to refer out (28.9%).

In binomial logistic regression models for reported reductions in cervical cancer screening tests, gender and ethnicity were sequentially removed due to p-values exceeding 0.10. See *Table 3*, Panel A, for table displaying logistic regression findings and *Figure 2*, Panels A–E, for forest plots depicting adjusted odds ratios and 95% confidence intervals for variables associated with odds of reporting reduced cervical cancer screening in 2021 compared with before the COVID-19 pandemic. In the final model, older age, non-White race, family or internal medicine physician specialty, and practicing in a community health/safety net setting were significantly associated with the likelihood of reporting reductions in cervical cancer screenings in 2021 compared to before the COVID-19 pandemic. Older age was associated with reported reductions in cervical cancer screening (p<0.001). Compared with clinicians over age 60, younger age groups were significantly less likely to report reduced cervical cancer screenings (<40 [aOR = 0.47, 95% CI: 0.33–0.66, p=0.000], 40–59 [aOR = 0.64, 95% CI: 0.46–0.90, p=0.009], 50–59 [aOR = 0.70, 95% CI: 0.51–0.97, p=0.029]). Race was marginally associated with reported reduced cervical cancer screening (p=0.085). Compared with White clinicians, Black (aOR 1.65, 95% CI: 0.96–2.84, p=0.070) and mixed race/other clinicians (aOR = 1.69, 95% CI:

**Table 1.** Demographic and practice characteristics for the full sample of respondents, and sub-groups of colposcopists and qualitative interview participants.

| Variable | Total sample | | | Colposcopist sub-group | | | Qualitative interview sub-group | | |
|---|---|---|---|---|---|---|---|---|---|
| | N | % | Valid N | N | % | Valid N | N | % | Valid N |
| **Clinician characteristics** | | | | | | | | | |
| Age | | | 1250 | | | 674 | | | 52 |
| Less than 40 | 277 | 22.2 | | 105 | 15.6 | | 8 | 15.4 | |
| 40–49 | 313 | 25.0 | | 157 | 23.3 | | 17 | 32.7 | |
| 50–59 | 344 | 27.5 | | 216 | 32.0 | | 12 | 23.1 | |
| 60+ | 316 | 25.3 | | 196 | 29.0 | | 15 | 28.8 | |
| Gender identity | | | 1250 | | | 674 | | | 55 |
| Female (includes transgender/gender non-binary and other)* | 934 | 74.7 | | 474 | 70.3 | | 38 | 69.1 | |
| Male | 316 | 25.3 | | 200 | 29.7 | | 17 | 30.9 | |
| Race | | | 1245 | | | 671 | | | 55 |
| Asian | 151 | 12.1 | | 71 | 10.6 | | 10 | 18.5 | |
| Black/African American | 61 | 4.9 | | 30 | 4.5 | | 1 | 1.9 | |
| Mixed race/other† | 64 | 5.1 | | 38 | 5.7 | | 5 | 9.3 | |
| White | 969 | 77.8 | | 532 | 79.3 | | 38 | 70.4 | |
| Hispanic/Latinx | 101 | 8.1 | 1247 | 51 | 7.6 | 672 | 2 | 3.6 | 55 |
| Clinician type (training and specialty) | | | 1250 | | | 674 | | | 55 |
| APP (total) | 609 | 48.7 | | 244 | 36.2 | 674 | 21 | 38.2 | |
| Sub-groups:<br>Nurse Practitioner<br>Certified Nurse Midwife<br>Physician Assistant | 521<br>71<br>11 | 85.6<br>11.7<br>1.8 | | 202<br>36<br>6 | 82.8<br>14.8<br>2.5 | | 19<br>1<br>1 | 90.4<br>4.8<br>4.8 | |
| MD/DO OB/GYN | 332 | 26.6 | | 317 | 47.0 | | 16 | 29.1 | |
| MD/DO family medicine | 200 | 16.0 | | 93 | 13.8 | | 12 | 21.8 | |
| MD/DO internal medicine | 109 | 8.7 | | 20 | 3.0 | | 6 | 10.9 | |
| **Practice characteristics, patterns, and behaviors** | | | | | | | | | |
| Type of practice | | | 1251 | | | 675 | | | 51 |
| Academic medical center | 154 | 12.3 | | 88 | 13.0 | | 4 | 7.8 | |
| Hospital-based practice (includes 'other') | 169 | 13.5 | | 85 | 12.6 | | 7 | 13.7 | |
| Private practice/group practice | 678 | 54.2 | | 395 | 58.5 | | 27 | 52.9 | |
| FQHC/community health center/planned parenthood or public health department | 250 | 20.0 | | 107 | 15.9 | | 13 | 25.5 | |
| US region | | | 1251 | | | 675 | | | 55 |
| Northeast | 230 | 18.4 | | 122 | 18.1 | | 8 | 14.5 | |
| South | 361 | 28.9 | | 190 | 28.1 | | 10 | 18.2 | |
| Midwest | 271 | 21.7 | | 121 | 17.9 | | 8 | 14.5 | |
| West | 277 | 22.1 | | 152 | 22.5 | | 5 | 9.1 | |
| Non-responders | 112 | 9.0 | | 90 | 13.3 | | 24 | 43.6 | |

*Due to small numbers, transgender/non-binary/other were unable to be analyzed as their own category. They were assigned to female for analyses because female was the most common response. No difference was noted when grouped with male.

†Due to small numbers, the following categories were combined: mixed race n=36, Hawaiian/AAPI n=3, American Indian/Alaska Native n=3, other n=22.

**Table 2.** COVID-19 and pandemic-related responses for the full sample of respondents and for colposcopists.

| Variable | Total sample | | | Colposcopist sub-sample | | |
|---|---|---|---|---|---|---|
| | N | % | Valid N | N | % | Valid N |
| How has the pandemic affected your cervical cancer screening practice? (data collected March-July 2021) | | | 1246 | | | 672 |
| Doing <u>fewer</u> Pap/HPV/co-tests now than before the pandemic | 586 | 47.0 | | 295 | 43.9 | |
| Doing the <u>same</u> number of Pap/HPV/co-tests now than before the pandemic | 604 | 48.5 | | 345 | 51.3 | |
| Doing more Pap/HPV/co-tests now than before the pandemic | 56 | 4.5 | | 32 | 4.8 | |
| How has the pandemic affected your colposcopy practice? (data collected March-July 2021) | | | - | | | 671 |
| Doing <u>fewer</u> colposcopies now than before the pandemic | - | - | - | 296 | 44.1 | |
| Doing the <u>same</u> number of colposcopies now than before the pandemic | - | - | - | 352 | 52.5 | |
| Doing more colposcopies now than before the pandemic | - | - | - | 23 | 3.4 | |
| How has the pandemic affected the ability to provide LEEP in your practice? (data collected March-July 2021) | | | - | | | 667 |
| We provided LEEP to patients on site before COVID-19 and are still doing so with the <u>same</u> capacity | - | - | - | 341 | 51.1 | |
| We provided LEEP to patients on site before COVID-19 and are still doing so but with <u>reduced</u> capacity | - | - | - | 124 | 18.6 | |
| We provided LEEP to patients on site before COVID-19 but now are referring to another facility | - | - | - | 9 | 1.3 | |
| We have always referred to another facility for LEEP and continue to do so | | | | 193 | 28.9 | |

0.99–2.88, p=0.055) more frequently reported reduced screenings. Clinician type was significantly associated with odds of reporting reduced screening during the pandemic (p<0.001). Compared with OB/GYN physicians, reduced screening was more frequently reported by internal medicine (aOR = 2.59, 95% CI: 1.62–4.13, p<0.001) and family medicine physicians (aOR = 1.64, 95% CI: 1.14–2.36, p=0.008). Practice type was significantly associated with odds of reporting reduced screening during the pandemic (p=0.014). Compared with those in private practice, those practicing in community health/safety net settings more often reported reduced screening (aOR = 1.62, 95% CI: 1.17–2.23, p=0.003). As specified in the Methods section, the model was adjusted for provider region, despite its lack of significant association with changes in screening (p=0.391).

In models with the subset of colposcopists, ethnicity, race, and age were sequentially removed from models due to p-values exceeding 0.10. See *Table 3*, Panel B, for table displaying logistic regression findings and *Figure 3*, Panels A–D, for forest plots depicting adjusted odds ratios and 95% confidence intervals for variables associated with odds of reporting reduced colposcopies in 2021 compared with before the COVID-19 pandemic. Among colposcopists, male gender and internal medicine specialty were associated with odds of reporting fewer colposcopies during the pandemic. Males reported reduced colposcopies marginally more often than females (aOR = 1.46, 95% CI: 0.98–2.18, p=0.063), and internal medicine physicians more often reported significantly reduced colposcopies than OB/GYN physicians (aOR = 3.79, 95% CI: 1.33–10.80, p=0.013). US region was not associated with perceived colposcopy reduction (p=0.414). Similarly, although the overall association between practice type and perceived colposcopy reduction was not statistically significant (p=0.266), clinicians in community health/safety net settings reported reduced colposcopy more often than their peers in private practice (aOR = 1.59, 95% CI: 1.01–2.53, p=0.048).

## Qualitative interview findings

A subset of 55 clinicians participated in qualitative interviews. The demographic characteristics of the qualitative interview sub-sample resembled that of the full sample (*Table 1*); they were primarily White (70%), non-Hispanic (96%), and female (69%). More than one-third (38.5%) were APPs, 29% were OB/

**Table 3.** Final models for variables associated with odds of reporting reduced cervical cancer screenings (Panel A) and with odds of reporting reduced colposcopies (Panel B) in 2021* compared with before the COVID-19 pandemic.

**Panel A. Final model of clinician and practice characteristics associated with odds of reporting reduced cervical cancer screenings in 2021 compared with before the COVID-19 pandemic (N=1239). Using backward selection, the following variables sequentially fell out of the model (p>0.10): (1) gender, (2) ethnicity. (A priori we planned to retain clinician type, practice type, and region even when p>0.10.)**

| | Overall p | B | SE | Odds ratio | p | CI |
|---|---|---|---|---|---|---|
| Age | <0.001 | | | | | |
| <40 | | −0.77 | 0.18 | 0.47 | <0.001 | 0.33-0.66 |
| 40–59 | | −0.44 | 0.17 | 0.64 | 0.009 | 0.46-0.90 |
| 50–59 | | −0.35 | 0.16 | 0.70 | 0.029 | 0.51-0.97 |
| 60+ (ref) | | - | - | - | - | - |
| Race | 0.085 | | | | | |
| Mixed race/other | | 0.52 | 0.27 | 1.69 | 0.055 | 0.99–2.88 |
| Black/African American | | 0.50 | 0.28 | 1.65 | 0.070 | 0.96–2.84 |
| Asian | | 0.03 | 0.19 | 1.03 | 0.882 | 0.71–1.50 |
| White (ref) | | - | - | - | - | - |
| Region | 0.391 | | | | | |
| No response | | −0.38 | 0.25 | 0.69 | 0.123 | 0.42–1.11 |
| South | | 0.08 | 0.18 | 1.08 | 0.650 | 0.77–1.52 |
| Midwest | | −0.03 | 0.19 | 0.97 | 0.859 | 0.67–1.40 |
| West | | 0.05 | 0.19 | 1.05 | 0.795 | 0.73–1.51 |
| Northeast (ref) | | - | - | - | - | - |
| Clinician type | <0.001 | | | | | |
| AAP | | −0.03 | 0.15 | 0.97 | 0.846 | 0.72–1.31 |
| MD/DO Internal Med | | 0.95 | 0.24 | 2.59 | <0.001 | 1.62–4.13 |
| MD/DO Fam Med | | 0.49 | 0.19 | 1.64 | 0.008 | 1.14–2.36 |
| MD/DO OB/GYN (ref) | | - | - | - | - | - |
| Practice type | 0.014 | | | | | |
| Academic medical center | | 0.03 | 0.19 | 1.03 | 0.889 | 0.71–1.48 |
| Hospital-based practice | | −0.11 | 0.19 | 0.90 | 0.554 | 0.62–1.29 |
| Public health dept/ FQHC/community health center/planned parenthood | | 0.48 | 0.16 | 1.62 | 0.003 | 1.17–2.23 |
| Private practice/group practice (ref) | | - | - | - | - | - |

**Panel B. Final model of clinician and practice characteristics associated with odds of reporting reduced colposcopies in 2021 compared with before the COVID-19 pandemic for colposcopists only (N=669). Using backward selection, the following variables sequentially fell out of the model (p>0.10): (1) ethnicity, (2) race, (3) age. (A priori we planned to retain clinician type, practice type, and region even when p>0.10.)**

| | Overall p | B | SE | Odds ratio | p | CI |
|---|---|---|---|---|---|---|
| Gender | 0.063 | | | | | |
| Male | | 0.38 | 0.20 | 1.46 | 0.063 | 0.98–2.18 |
| Female (ref) | | - | - | - | - | - |
| Region | 0.414 | | | | | |
| No response | | 0.08 | 0.30 | 1.08 | 0.785 | 0.61–1.94 |
| South | | 0.43 | 0.24 | 1.54 | 0.077 | 0.96–2.47 |

*Table 3 continued on next page*

*Table 3 continued*

**Panel B. Final model of clinician and practice characteristics associated with odds of reporting reduced colposcopies in 2021 compared with before the COVID-19 pandemic for colposcopists only (N=669). Using backward selection, the following variables sequentially fell out of the model (p>0.10): (1) ethnicity, (2) race, (3) age. (A priori we planned to retain clinician type, practice type, and region even when p>0.10.)**

|  | Overall p | B | SE | Odds ratio | p | CI |
|---|---|---|---|---|---|---|
| Midwest |  | 0.27 | 0.27 | 1.30 | 0.320 | 0.77–2.20 |
| West |  | 0.33 | 0.25 | 1.39 | 0.200 | 0.84–2.28 |
| Northeast (ref) |  | - | - | - | - | - |
| Clinician type | 0.052 |  |  |  |  |  |
| Advanced practice professional |  | 0.26 | 0.20 | 1.30 | 0.197 | 0.87–1.92 |
| MD/DO Internal Med |  | 1.33 | 0.54 | 3.79 | 0.013 | 1.33–10.80 |
| MD/DO Fam Med |  | 0.00 | 0.25 | 1.00 | 0.995 | 0.62–1.62 |
| MD/DO OB/GYN (ref) |  | - | - | - | - | - |
| Practice type | .266 |  |  |  |  |  |
| Academic medical center |  | 0.15 | 0.25 | 1.16 | 0.554 | 0.71–1.88 |
| Hospital-based practice |  | 0.09 | 0.25 | 1.09 | 0.725 | 0.67–1.79 |
| Public health dept/ FQHC/community health center/planned parenthood |  | 0.47 | 0.24 | 1.59 | 0.048 | 1.01–2.53 |
| Private practice/group practice (ref) |  | - | - | - | - | - |

*Note. Data collected during the COVID-19 pandemic period of March 2021–July 2021, with participants asked to report whether they were doing 'fewer, the same number, or more Pap/HPV/co-tests' or 'colposcopies' 'now than before the pandemic'.

GYN physicians, and the remainder were internal (11%) and family medicine (22%) physicians; about half (47%) indicated they perform colposcopies.

*Table 4* illustrates themes described by clinicians related to perceived screening and colposcopy changes connected with the COVID-19 pandemic, along with exemplar quotes. Themes included reductions in screening, rebound to pre-pandemic levels, and tracking systems for patient follow-up. Nearly all clinicians described reductions in screening early in the pandemic. Sub-themes included closures of primary care services, prioritization of acute problems over well visits, prioritization of abnormal Pap test results over routine screening, patient fears of contracting COVID-19 if they visited a medical setting, and the shift to telehealth limiting in person services. One clinician stated: "My clinic was stopping…annual wellness exams for a 6-month time period. So, there were a lot of patients that were kind of put off during that time period" (APP, practice not specified). Another described prioritization of colposcopy visits based on the severity of the Pap result: "As soon as we were able to provide those services, we prioritized the visit based on the Pap result. So, high-grade had a high priority to come in for the colpo[scopy] before the low grade" (APP, private practice). Another described both the impact of telemedicine and prioritization of illness over wellness care: "At the beginning of the pandemic, we were exclusively telemedicine for a few months and then as we were opening up office visits, cervical cancer screening was not the highest priority. It was more so our chronic care patients" (MD/DO family medicine, safety net setting). Participants also described patient concerns: "They don't wanna come into the health department because they think we're full of COVID germs or something" (APP, safety net setting). Others noted that patient volumes had not recovered: "It is still a little less. I would say 80% now compared to before COVID" (MD/DO internal med, private practice).

In contrast, a few clinicians stated that services were never curtailed due to the pandemic, and several felt that screening had rebounded, or in some cases exceeded, pre-pandemic levels: "Last year, we had less patients coming back for physicals. With COVID, they weren't coming. I think people are catching up now. This year, we're seeing more volume. More patients coming for their annuals and their Pap smears; I think we are back up to the pre COVID volume" (MD/DO family medicine, private practice). Some described feeling inundated with cervical cancer screening: "I'm just the non-stop

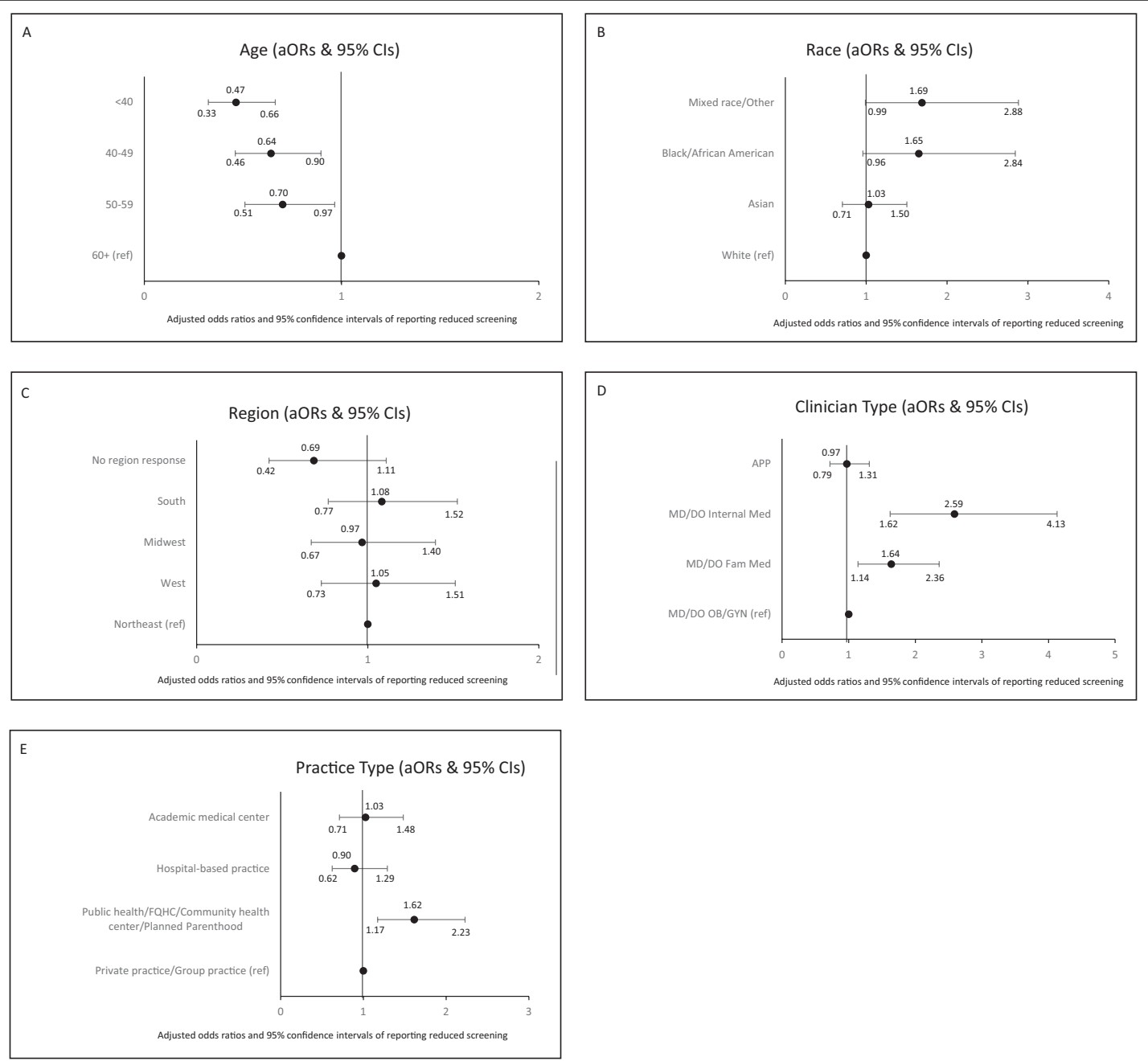

**Figure 2.** Forest plots depicting adjusted odds ratios and 95% confidence intervals for variables associated with odds of reporting reduced cervical cancer screening (N=1239) in 2021* compared with before the COVID-19 pandemic. Variables associated with odds of reporting reduced cervical cancer screening include Panel **A**: Age; **B**: Race; **C**: Region; **D**: Clinician Type; **E**: Practice Type. *Note. Data collected during the COVID-19 pandemic period of March 2021–July 2021, with participants asked to report whether they were doing 'fewer' or 'the same number or more Pap/HPV/co-tests now than before the pandemic'.

Pap clinic" (APP, safety net setting). Some clinicians noted that patients were less fearful of attending medical care after widespread vaccination.

When asked what prompted patients to return for screening, clinicians reported a range of practices related to patient outreach and tracking systems. Some clinicians reported using the electronic medical record to outreach to patients, though more reported using a combination of patient lists and outreach via staff phone calls: "We still maintained our rescreen list and our no-show list, and we recalled those patients" (APP, safety net setting). Concerningly, several clinicians did not believe

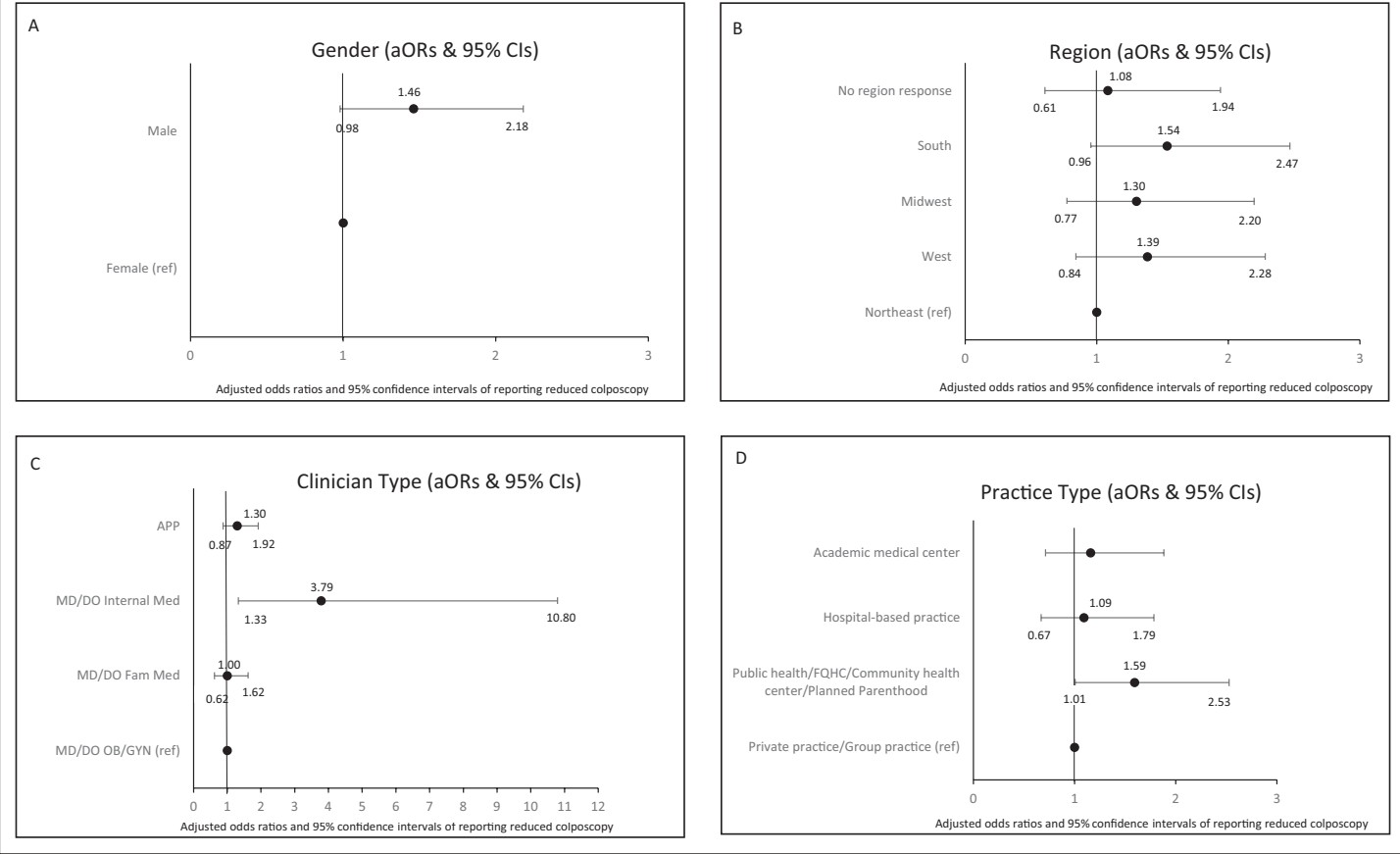

**Figure 3.** Forest plots depicting adjusted odds ratios and 95% confidence intervals for variables associated with odds of reporting reduced colposcopies (N=669), in 2021* compared with before the COVID-19 pandemic. Variables associated with odds of reporting reduced colposcopies include Panel **A**: Gender; **B**: Region; **C**: Clinician Type; **D**: Practice Type. *Note. Data collected during the COVID-19 pandemic period of March 2021– July 2021, with participants asked to report whether they were doing 'fewer' versus 'the same number or more colposcopies now than before the pandemic'.

they had a reliable system for tracking patients who had missed screenings: "Once they cancel or don't show up, they do lose the follow up" (MD/DO OB/Gyn, private practice), or were unaware of what system their staff might be using to track missed screenings. Some indicated that patients were responsible for making and rescheduling appointments: "We just sent people notices saying 'she's [the doctor's] back in the office. Hook up if you need to'" (APP, private practice).

## Discussion

We examined quantitative and qualitative data related to the perceived impact of the COVID-19 pandemic on cervical cancer screening and abnormal follow-ups in a large, national sample of clinicians who provide these services. Over 1 year into the pandemic, almost half of clinicians still reported a reduction in cervical cancer screenings (47%) and colposcopies (44%) as compared to the time preceding the pandemic. These findings are salient because at the time of our study most lockdown restrictions had been lifted and healthcare institutions had returned to near pre-pandemic level of operations, albeit with masking and social distancing in place. Qualitative themes indicated that some clinicians' patient screening volumes were similar to or exceeding pre-pandemic levels as they caught up on backlogs, while others were slower to return to pre-pandemic cervical cancer screening and management. Many faced challenges associated with follow-up and tracking systems for overdue patients.

Clinician age, clinician type, and practice setting were significantly associated with reports of performing fewer cervical cancer screenings during the pandemic, with similar but marginal associations

**Table 4.** Themes demonstrating screening and colposcopy changes during COVID-19 pandemic, with related quotes (interviews conducted June-December 2021).

| Reduced screening | Example quotes |
|---|---|
| Closure of primary care services | "Clinics were shut down... Staff were furloughed. So, a lot of things that were going on. So, there was a pause for a time in the clinic. I don't know, maybe for a few months." (APP, private practice)<br>"Screenings, wellness visits were almost at a standstill for almost 6 months out of the 15 months we've had COVID for. And we started back three months ago, where we started the wellness and screening test again. Then I don't know how long we can continue to do it with the second and the fourth waves." (MD/DO int med, private practice) |
| Prioritization of COVID-related or urgent services | "Screening was the least of the problems with clinics and hospitals and urgent cares full of the pandemic. So it kind of took a back seat for a long time, which was helpful, you know, treating the disease, but not helpful in the screening world." (MD/DO Ob/Gyn, academic medical center)<br>"Most people have had their wellness exam delayed because we're seeing patients – we will give priority to patients with problems." (APP, private practice) |
| De-prioritization of screening/ focus on abnormal follow-up | "Knowing that pap smears and dysplasia don't progress fast, we did postpone and delay for a period of time, it was three to six months from what I recall during the early days of the pandemic." (MD/DO Ob/Gyn, academic medical center)<br>"For the people who needed them, they were getting them throughout, not those, not those first three months unless it was a CIN-2 or higher, but if it was a CIN-1 we were like, let's just wait a few more months… see what's going on with COVID at that point." (APP, private practice) |
| Patient fear of attending medical care | "You can only do so much; patients are scared to come to doctors' offices for screening test. If it's not urgent or not emergent, very few patients actually want to go to a healthcare setting." (MD/DO int med, private practice)<br>"We still have a significant number who are not comfortable with an in-office visit unless they need something like their birth control pill or something else." (APP, private practice) |
| Telehealth reduces screening | "We do telehealth but obviously – we can possibly do STD health and STD screenings on telehealth, but aside from that unless there's some symptomatic issues that we can try to take care of over the telehealth platform, we – there's just been a lot less women coming into the office over the last year due to COVID." (MD/DO fam med, private practice)<br>"I'm sure the Pap smear volume is much more you know, reduced due to the Telemedicine. And so that's one effect of COVID. It's reduced COVID, yeah it reduced the Pap smear." (MD/DO int med, private practice) |
| Patient volumes remain below pre-COVID levels | "The rate of visits and doing their regular checkups, not only cancer screening has dropped significantly, more than 75% during the year of COVID. I'm sure that the results of these problems will arise in the next few years, if not, decades; We're almost back 80%, so we're still 20% lower than…usually we see at this time of the year." (MD/DO fam med, private practice)<br>"Getting patients into the office has been challenging because we were doing more telemedicine visits, not as many in-person visits, and the perceived need of preventative care had changed during COVID as well." (MD/DO fam med, safety net setting) |

| Rebound in screening to pre-pandemic levels | Example quotes |
|---|---|
| Offered cervical cancer screening and colposcopy throughout pandemic | "Absolutely no impact because we were open throughout, and I was doing full scope everything, because we just… staggered patients. And so we were just working longer hours with time in between patients and cleaning up and so absolutely no impact." (MD/DO Ob/Gyn, safety net setting)<br>"There was an executive order by the governor who said there will be no elective surgeries done." And, you know, there are, there are people that said, "Oh, well, colposcopy is like an elective office surgery, theoretically." Or a LEEP is – "And I said, screw that. I created a form saying in my opinion, you know, delaying this biopsy or delaying this treatment may cause whatever abnormality [to] worsen." (MD/DO Ob/Gyn, private practice) |
| Patients willing to come in once services were restored | "Now we do have patients come in and… we're almost back to regular business except for masking. So, I would say now, it's just no different than it was before." (APP, safety net setting)<br>"Instead of doing 10 annuals a day, probably three annuals a day throughout COVID before vaccination, then after vaccination, people started coming in in droves for their annuals." (APP, private practice) |
| Increased screening to compensate for patient backlogs | "Literally, from June [2021], it started backed up open full force. I have more patients that I had before." (MD/DO Ob/Gyn, practice type not specified)<br>"I will say that coming back off a furlough we might, not only my schedules, our schedules are packed, so we were kind of making up for lost time." (APP, private practice)<br>"So, we do outreach and do pap clinics on Saturdays to help folks get caught up." (APP, safety net setting) |
| Patients no longer afraid of COVID | "They became more complacent. So, now, they are not as afraid.<br>So, well, that may have helped the screening process, but it may not help the fact that they are going to get infected." (MD/DO int med, private practice)<br>"You know we are in Alabama. People here don't think we have COVID even though they die with the same numbers as everybody else." (APP, academic medical center) |

*Table 4 continued on next page*

*Table 4 continued*

| Patient tracking and outreach | Example quotes |
|---|---|
| Active follow-up system | "We can run lists based on who needs annual wellness visits, who needs mammograms, who needs Pap smears, who needs diabetic follow up, -you name it, we can run the list. And so, since we opened back up …We have people that call [patients]. …that's their job." (APP, practice type not specified)<br>"I think we've got dedicated staff who actually run through the charts and see who has missed their well-woman exams, and they either make a phone call or send a postcard." (MD/DO fam med, private practice) |
| Limitations of tracking/ follow-up systems | [There was a list] "but that's gone by the wayside. I don't know how they did that. They were like, 'Oh, we're going to call all the old people [previous patients]…'. I don't know if that worked." (APP, practice type not specified)<br>"We're trying to call, to arrange for phone calls to get them back, but of course, it's very time, labor consuming process to go through each patient see when was the last time they were here. The good thing is we start implementing the year before COVID, the Epic, myChart portal. You can adjust it to send the patient's notification when their checkups are due. From there, of course at least 50% of our population, they are not tech savvy. They don't check that on regular basis, but we try to reach as much as we can. It's definitely a challenging issue based on human factor and administrative factors, too." (MD/DO fam med, private practice)<br>"We are being proactive and our EMR will send the messages and reminders. But specifically, for patients who didn't come last year, we do not have a system." (MD/DO fam med, private practice)<br>"It's up to the patient to kind of know that they needed a Pap test or a colposcopy or something like that and then make an appointment." (MD/DO Ob/Gyn, practice type not specified) |
| EMR facilitates tracking | "We have a portal that our patients have access to. Once we knew that [COVID-19] vaccination was really prevalent in our area… then we, through the portal, sent out a mass announcement to everyone." (MD/DO Ob/Gyn, private practice)<br>"In my PracticeSuite, I have this alert system. So, somebody who has an abnormal pap… I put an alert in... and then it will pop up, and then I just call them." (APP, safety net setting) |

with race. Clinicians over age 60 more often reported fewer screenings than younger clinicians. This could indicate that older clinicians were more cautious in returning to in-person care such as cervical cancer screenings, perhaps due to their own health concerns over age-related susceptibility to severe COVID-19 complications (*Dessie and Zewotir, 2021*). Internal and family medicine physicians more often reported reduced cervical cancer screening compared with OB/GYN physicians. The need for internal and family medicine physicians to care for COVID-19 patients and other acute health issues may have impacted their ability to provide preventive or well care (*Turner et al., 2022*), including cervical cancer screenings (*Kim et al., 2022*). We also found that clinicians in safety net settings like community health centers and health departments more often reported reduced screenings *and* colposcopies during the pandemic. Concerningly, this may indicate worsening disparities in cancer prevention care in settings that serve patients with the highest cervical cancer rates: lower resourced and historically marginalized communities (*Moss et al., 2022*). These findings are supported by recent literature indicating that federally qualified health center settings suffered staffing losses and other challenges during the pandemic which led to reductions in cancer screenings due to postponement of preventive care (*Fisher-Borne et al., 2021*). Finally, we found that mixed race/other and Black clinicians had a marginally associated, but higher likelihood of reporting reduced cervical cancer screenings compared to White clinicians, independent of other factors such as age, gender, region, medical specialty, and practice setting. There is a paucity of literature on differences in screening practices by clinician race/ethnicity, and additional research would be helpful to further explore these findings.

When focusing only on clinicians who perform colposcopy, we found that internal medicine physicians more often reported reduced colposcopies compared to other specialties. The need for internal medicine physicians to address more acute patient issues may have contributed to reported reductions in both screening and colposcopy. We also found a marginal association indicating that male clinicians more often reported reduced colposcopy than female clinicians. Some practices and/or states have varying guidelines around chaperone requirements during pelvic exams and procedures, which may have impacted the ability to perform colposcopy. This finding warrants further exploration. Our data also indicated reductions in providing office-based treatment for cervical precancer (e.g., LEEP). In some offices the extra staffing and cleaning associated with performing a LEEP may have led to reduced availability or a need to refer out to another facility during the pandemic. Together these findings highlight perceived reductions in cervical cancer preventive care overall, but were more prevalent among certain specialties and practice settings. Further research is needed to confirm and explore these findings.

Qualitative interview findings provide insight into factors that contributed to screening reductions, as well as the trajectory of care in different phases of the pandemic. Participants highlighted strategies used to mitigate the impact of clinic closures or reduced capacity during the pandemic, including prioritization of seeing patients with high-grade abnormal results during periods when care was restricted (*Wentzensen et al., 2021*; *Masson, 2021*). As COVID-19-related restrictions were lifted, clinicians described implementing extended weekday hours or weekend screening only clinics as they caught up on screenings and compensated for social distancing/reduced capacity restrictions. The introduction or expansion of telehealth during the pandemic was described as helpful with addressing acute concerns while minimizing infection risk. Consistent with prior literature, however, clinicians reported that telemedicine hindered cervical cancer screening because patients were not physically attending clinic where opportunistic screening could occur (*Price et al., 2022*).

Return to screening and the ability to recall overdue patients varied greatly. Some clinicians described an overabundance of patients returning to clinical care, while others described challenges with reaching patients who were overdue for screening or follow-up care for abnormal results. Several described staffing shortages that impacted screening, consistent with rapid turnover and a reduction in the healthcare workforce since the beginning of the pandemic (*Massad, 2022*; *Falatah, 2021*). While some clinicians reported having a formal tracking system to determine which patients need screening and follow-up, others emphasized limitations of tracking and outreach, such as electronic medical record limitations and the time-intensive burden of outreach for staff. Several clinicians were unaware of whether they had a tracking system for overdue screenings, or how patient recall was implemented.

Together these findings highlight perceived reductions in cervical cancer preventive care throughout the cancer prevention continuum of screening, diagnosis via colposcopy, and treatment via LEEP. If not addressed, reductions in cancer prevention services could lead to increased cancer incidence in the future. It is well known that cancer screenings in the US decreased dramatically at the height of the pandemic (*Chen et al., 2021*; *Poljak et al., 2021*; *Amram et al., 2022*; *Smith and Perkins, 2022*), with cervical cancer screening rates dropping in 2020 compared with previous years (*Miller et al., 2021*; *Mayo et al., 2021*). As the pandemic progressed, cancer screening rates started to rebound (*Chen et al., 2021*; *McBain et al., 2021*), but our findings highlight challenges that still exist for cervical cancer screening and colposcopy. Interestingly, similar challenges in getting back to screening and treatment have been observed in low- and middle-income countries, as well (*Villain et al., 2021*). In the present study, reductions in screening and follow-up were reported overall, but were more prevalent among internal medicine physicians and community health/safety net settings of care. Because current cervical cancer screening requires an in-person exam and sometimes a chaperone, it is relatively labor-intensive process for primary care clinicians compared to other screenings that require only laboratory orders or referrals. The ability for patients to self-collect vaginal specimens for HPV testing could be one method of reducing workforce burden and increasing access to cervical cancer screening (*Fuzzell et al., 2021*). Some countries currently use self-sampling within larger population-based screening programs to reach individuals who have barriers to screening (*Arbyn et al., 2014*; *Serrano et al., 2022*; *Inturrisi et al., 2021*).

This study has several inherent strengths and weaknesses. As noted in our prior work (*Vadaparampil et al., 2023*), this sample includes both primary care and OB/GYN physicians, and APPs who conduct cervical cancer screening across various practice settings and regions of the US. We worked with both ASCCP and NPWH in order to ensure sufficient samples of both physicians who perform colposcopy and APPs, who are often overlooked in clinician surveys. Survey data were supplemented with more in-depth exploration via qualitative interviews; the large sample of survey respondents coupled with a relatively large qualitative sample eliciting rich responses are core strengths of this study. Although not originally targeted at assessing changes in screening and colposcopy during the pandemic, we included items to assess provider perceptions of these impacts because of early literature suggesting a drop in screenings with slow rebound (*Chen et al., 2021*; *McBain et al., 2021*). Finally, to our knowledge, this is the first US report detailing changes in colposcopy practices during the pandemic, a unique addition to the literature. These strengths are tempered by some limitations. The majority of the sample were White and non-Hispanic, although these characteristics reflect characteristics of healthcare providers in the US (*United States Census Bureau, 2022*). Next, 9% of the full sample did not respond to geographic location items, thus this gap in data may have limited our ability to detect

differences surrounding region- and pandemic-related differences in screening and colposcopy. Finally, self-report surveys have inherent biases and may not be actual representations of screening and colposcopy practices that could be ascertained via medical record or claims databases.

These findings highlight that nearly half of clinicians reported performing fewer cervical cancer screenings and colposcopies compared to before the pandemic. This is particularly concerning as this survey occurred more than 1 year into the pandemic, after lockdowns had been lifted and when widespread vaccination was available. Persistent reductions in screening and colposcopy could lead to increases in cervical cancer incidence in the near future. Additional research should track whether cervical cancer screening services have continued to recover, and whether inequities in recovery exist that could worsen cervical cancer disparities.

## Additional information

### Funding

| Funder | Grant reference number | Author |
|---|---|---|
| American Cancer Society | | Susan T Vadaparampil |

The funders had no role in study design, data collection and interpretation, or the decision to submit the work for publication.

### Author contributions

Lindsay Fuzzell, Data curation, Formal analysis, Visualization, Writing - original draft, Writing – review and editing; Naomi C Brownstein, Data curation, Supervision, Investigation, Visualization, Methodology, Writing – review and editing; Holly B Fontenot, Formal analysis, Investigation, Methodology, Writing – review and editing; Paige W Lake, Ashley Whitmer, Writing – review and editing; Alexandra Michel, Sarah L Rossi, Formal analysis, Writing – review and editing; McKenzie McIntyre, Data curation, Writing – review and editing; Susan T Vadaparampil, Rebecca B Perkins, Conceptualization, Resources, Software, Supervision, Funding acquisition, Validation, Investigation, Methodology, Writing – review and editing

### Author ORCIDs

Lindsay Fuzzell http://orcid.org/0000-0001-9688-5365
Paige W Lake https://orcid.org/0000-0002-5591-6417

### Ethics

Human subjects: This study was approved by Moffitt Cancer Center's Scientific Review Committee and was reviewed by an Institutional Review Board. The study was given exempt determination by Moffitt's IRB, Advarra (MCC #20048), and Boston University's IRB (BMC IRB# H-41533). Informed consent, and consent to publish, was obtained from all participants.

### Decision letter and Author response

Decision letter https://doi.org/10.7554/eLife.85682.sa1
Author response https://doi.org/10.7554/eLife.85682.sa2

## Additional files

### Supplementary files
• MDAR checklist

### Data availability

Full human subjects data are unavailable via a data repository due to confidentiality concerns. A limited dataset may be made available upon reasonable request from other academic researchers and requests should be submitted via email to the corresponding author and will be approved on a case by case basis by study PIs and the institutional SRC and IRB. SPSS version 26 was used to analyze data. SPSS code has been made available.

The following dataset was generated:

| Author(s) | Year | Dataset title | Dataset URL | Database and Identifier |
|-----------|------|---------------|-------------|-------------------------|
| Fuzzell L | 2023 | CC PROGRESS pandemic analysis syntax | https://doi.org/10.7910/DVN/XD8YE9 | Harvard Dataverse, 10.7910/DVN/XD8YE9 |

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
