## [Editor Report]

This important work provides evidence regarding the impact of the COVID-19 pandemic on cervical cancer screening and precancer treatments in the USA. As there are few screening registries, the study provides solid evidence using a survey of health providers' impressions to assess whether cervical cancer screening services declined during the pandemic. The work will be of interest to public health professionals working in cancer prevention.

---

## [Decision Letter]

**Decision letter after peer review:**

Thank you for submitting your article "A cross sectional survey examining the association of clinician characteristics with perceived changes in cervical cancer screening and colposcopy practice during the COVID-19 pandemic" for consideration by *eLife*. Your article has been reviewed by 3 peer reviewers, and the evaluation has been overseen by a Reviewing Editor and a Senior Editor. The following individuals involved in the review of your submission have agreed to reveal their identity: Richard Shellenberger (Reviewer #2); Patricia Villain (Reviewer #3).

As is customary in *eLife*, the reviewers have discussed their critiques with one another. What follows below is the Reviewing Editor's edited compilation of the essential and ancillary points provided by reviewers in their critiques and in their interaction post-review. Please submit a revised version that addresses these concerns directly. Although we expect that you will address these comments in your response letter, we also need to see the corresponding revision clearly marked in the text of the manuscript. Some of the reviewers' comments may seem to be simple queries or challenges that do not prompt revisions to the text. Please keep in mind, however, that readers may have the same perspective as the reviewers. Therefore, it is essential that you attempt to amend or expand the text to clarify the narrative accordingly.

Essential revisions:

1) A more detailed description of the "parent" survey questionnaire instrument and its purpose would be warranted.

2) Nuance results regarding race as a significant predictor as confidence intervals overlap with the null hypothesis.

3) Discuss potential biases in using a survey to gather this data.

4) Provide more context for an international audience regarding the impact and public health measures of the COVID-19 pandemic in the USA during the study period.

*Reviewer #1 (Recommendations for the authors):*

It would be helpful if the survey and interview instruments were provided.

A general description of the "parent" survey questionnaire instrument and its purpose would be helpful.

Comments specific to quantitative analysis (association between clinician characteristics and perceived changes in screening and colposcopy):

– A figure of participant enrollment and selection of the analytic sample is provided but not referred to in the manuscript text.

– Table 1 is a descriptive table of the three related study populations, as colposcopists and interviewed clinicians each constitute a subgroup of the total sample. However, the authors compared their characteristics, instead of comparing the characteristics of colposcopists to non-colposcopists for example. Can the authors comment on this?

– I agree with the authors' decision to force clinician type, practice type, and geographic region into the stepwise logistic regression models, but for the other variables (such as age, race, and ethnicity) the rationale for considering these as important determinants of screening and colposcopy practices during the pandemic is not presented.

– A simple graphical presentation of the three questions (impact on cervical cancer screening practice, colposcopy practice, ability to provide LEEP) by the more relevant variables (clinician type, practice type, and geographic region) would have been informative.

– Tables 3 (reduced screenings) and 4 (reduced colposcopies) could be combined, keeping only the odds ratios and confidence intervals columns (removing the other redundant columns).

Comments specific to qualitative analysis (how clinicians experienced pandemic-related changes in screening and colposcopy):

– Similar to the survey, the included qualitative interview items are part of a larger interview guide, of which details are lacking both on the overall and included interview items.

– How much overlap was there in the quantitative and qualitative components of the study? For example, between one of the three themes identified, the reductions in screening, and the survey responses.

*Reviewer #2 (Recommendations for the authors):*

Overall, I commend you highly for this important work. I am very impressed with the magnitude of this survey and the amount of demographic data acquired.

I would recommend nuancing the finding of Black and Asian providers as having statistically significant reductions in cervical cancer screening since the confidence intervals cross one in both instances for these. OR I would also add in some discussion about the inherent weaknesses in a survey study as well as potential biases which may exist. I would also take out the comment about regional differences in the way the pandemic was handled in different geographic regions in the US unless you have a statistic to support this statement. I understand your passion, but this statement shows some biases by the authors which may be not well received in certain areas and take away from the importance of your work.

Thank you for this well-done survey study. Your diligence and attention to detail in the statistical analysis are noteworthy. This is truly valuable public health information and a significant contribution to the field of cancer screening.

*Reviewer #3 (Recommendations for the authors):*

Because the study was conducted in 2021, some questions/points should be addressed to increase the overall impact of the work:

1) Highlight the context of the study: what was the situation in terms of the pandemic in the US when the study was conducted? This should be put in front in the introduction and results should be discussed regarding this context;

2) Results should be discussed in the light of what is already known/foreseen in terms of covid impact on cervical cancer screening and on cervical cancer;

3) It would be interesting to discuss these results i.e. obtained from a high-income country, compared to what was reported from low- and medium-income countries e.g. in terms of disruption of services, active follow-up of patients (https://doi.org/10.1002/ijc.33500);

4) Would results gained from this study be beneficial to improve cervical cancer screening or/and to health system responses to the pandemic?

5) Results of the survey suggest that inequality of cervical cancer screening access might have increased due to covid pandemic; do the authors have any comment on it?

The discussion should be revised to address the points listed above. I suggest also highlighting the pandemic context in the US when the study was conducted.

---

## [Author Response]

Essential revisions:1) A more detailed description of the "parent" survey questionnaire instrument and its purpose would be warranted.

We have added a slightly more detailed description of the survey and its main purpose (Survey Content and Study Variables, pg. 4).

“The survey covered several areas related to cervical cancer screening practices and management of abnormal screening results, including presentation of vignettes focused on screening intervals, management or treatment, and screening exit or continuation in relation to 2019 ASCCP risk-based management guidelines adoption, as well as a sub-set of items for clinicians who perform colposcopy. There were also items related to HPV self-sampling, as well as the impact of the COVID-19 pandemic on screening and follow-up (which is the focus of the present manuscript).”

We now also cite the recently published manuscript on pgs. 3 and 4. (Vadaparampil, S. T., Fuzzell, L. N., Brownstein, N. C., Fontenot, H. B., Lake, P., Michel, A., … and Perkins, R. B. (2023). A cross‐sectional survey examining clinician characteristics, practices, and attitudes associated with adoption of the 2019 American Society for Colposcopy and Cervical Pathology risk‐based management consensus guidelines. Cancer.)

2) Nuance results regarding race as a significant predictor as confidence intervals overlap with the null hypothesis.

Thank you for the suggestion. In terms of the associations of race with perceived reduced screenings (and gender with reduced colposcopies), we have tempered the language in the discussion for findings where confidence intervals cross 1 and/or where p-values are between.05 and.10 (for race in the screening model, see pgs. 11 and 12; for gender in the colposcopy mode, see pg. 12) and discuss them with language indicating “marginal” associations.

We also now explain on pg. 7 why we utilize a significance value of.10, which relates to the confidence intervals that cross 1:

“Given the few studies that have explored factors associated with clinician perspectives of changes in cervical cancer screenings and colposcopies during the pandemic, we selected a value for inclusion and significance of.10. This strikes a balance between the commonly accepted method of using the AIC (Akaike's Information Criterion, which implicitly assumes a significance level of 0.157), and the often-used significance level of 0.05.”

We now describe the choice of 0.10 in the text. However, we acknowledge that by using 0.10 as a significance level, some 95% confidence intervals for factors we consider significant cross 1. We have tempered language in the discussion for findings with p-values between 0.05 and 0.10.

3) Discuss potential biases in using a survey to gather this data.

We have added an acknowledgement of the inherent biases in survey data collection on pg. 13 of the discussion.

“self-report surveys have inherent biases and may not be actual representations of screening and colposcopy practices that could be ascertained via medical record or claims databases.”

4) Provide more context for an international audience regarding the impact and public health measures of the COVID-19 pandemic in the USA during the study period.

Data were collected between March-August 2021 (surveys) and June-December 2021 (interviews). For context, the COVID-19 vaccine became available to healthcare providers in the US in early 2021. The US general public had widespread access to vaccination beginning in the summer of 2021. By the fall of 2021, the pandemic appeared to be less acute in the US, with healthcare organizations attempting to resume normal operations through the end of the year. Masking, social distancing and reduced capacities indoors, and enhanced cleaning procedures were public health practices in place with varying levels of intensity across the US at this time. This information is now included on pg. 4 under Methods, Participant recruitment.

Reviewer #1 (Recommendations for the authors):It would be helpful if the survey and interview instruments were provided.

Thank you for the suggestion. We now include a more detailed description of the quantitative survey on pg. 4:

“The survey covered several areas related to cervical cancer screening practices and management of abnormal screening results, including presentation of vignettes focused on screening intervals, management or treatment, and screening exit or continuation in relation to 2019 ASCCP risk-based management guidelines adoption, as well as a sub-set of items for clinicians who perform colposcopy. There were also items related to HPV self-sampling, as well as the impact of the COVID-19 pandemic on screening and follow-up (which is the focus of the present manuscript).”

We also include a detailed description of the qualitative interview guide on pg. 6:

“The final interview guide included in depth exploration of cervical cancer screening and management items explored in the quantitative survey. We more deeply explored screening practices (barriers and facilitators to screening for each clinician’s patient panel), adherence to 2019 ASCCP guidelines (how clinicians assess if patients are due for screening, type of screening test used, screening interval used and reasoning) barriers to adoption of ASCCP guidelines, HPV self-sampling (benefits and concerns), and the impact of the pandemic on screening and management practices. Additionally, there was a sub-set of questions for colposcopists (on colposcopy training, LEEP self-performance versus referral, biopsy location).”

The quantitative survey measure and qualitative interview guide are available upon request.

A general description of the "parent" survey questionnaire instrument and its purpose would be helpful.

We have now added a more detailed description of the survey and its main purpose (see Survey Content and Study Variables, pg. 4).

“The survey covered several areas related to cervical cancer screening practices and management of abnormal screening results, including presentation of vignettes focused on screening intervals, management or treatment, and screening exit or continuation in relation to 2019 ASCCP risk-based management guidelines adoption, as well as a sub-set of items for clinicians who perform colposcopy. There were also items related to HPV self-sampling, as well as the impact of the COVID-19 pandemic on screening and follow-up (which is the focus of the present manuscript).”

We now also cite that recently published manuscript on pgs. 3 and 4.

Comments specific to quantitative analysis (association between clinician characteristics and perceived changes in screening and colposcopy):– A figure of participant enrollment and selection of the analytic sample is provided but not referred to in the manuscript text.

We now reference the figure on pg. 7 in the first paragraph of the Results.

– Table 1 is a descriptive table of the three related study populations, as colposcopists and interviewed clinicians each constitute a subgroup of the total sample. However, the authors compared their characteristics, instead of comparing the characteristics of colposcopists to non-colposcopists for example. Can the authors comment on this?

Thank you for your question. We report on the groups that we specifically designed our mixed methods study around. In addition to the full quantitative study population, we conducted interviews with a sub-sample of participants who completed surveys, and asked a sub-set of questions to colposcopists around their experience with colposcopy and LEEP procedures during the pandemic. Thus, we include descriptive statistics of each population or sub-population of interest, to correspond to the components of the study on which we report. We did not elect to report descriptive statistics on their counterparts, because these were not sub-groups of interest. (For example, we did not think it would be useful to present statistics on the portion of the sample who completed the quantitative survey, but not the qualitative survey.) Table 1 is purely descriptive, and no formal statistical tests were conducted to compare these nested populations.

– I agree with the authors' decision to force clinician type, practice type, and geographic region into the stepwise logistic regression models, but for the other variables (such as age, race, and ethnicity) the rationale for considering these as important determinants of screening and colposcopy practices during the pandemic is not presented.

Past literature has indicated associations exist between clinician characteristics and cervical cancer screening practices. We now cite this work in the method when discussing inclusion of variables in regression models (pg. 6, Analytic plan, Quantitative analyses). Although we did not these characteristics into the model, we did initially include these variables to assess possible associations.

– A simple graphical presentation of the three questions (impact on cervical cancer screening practice, colposcopy practice, ability to provide LEEP) by the more relevant variables (clinician type, practice type, and geographic region) would have been informative.

Thank you for your suggestion. We now include figures displaying forest plots of adjusted odds ratios and 95% confidence intervals for variables associated with screening and colposcopy outcomes.

– Tables 3 (reduced screenings) and 4 (reduced colposcopies) could be combined, keeping only the odds ratios and confidence intervals columns (removing the other redundant columns).

Thank you for the suggestion. We have combined tables 3 and 4, which is now labelled with panels A and B. We have elected to retain the B, SE, and p value columns as we believe these are informative for the statistics-minded reader. Tables were reviewed for consistency in decimal places and we now round to three decimal places for p values, but two decimal places for all other number in the table.

Comments specific to qualitative analysis (how clinicians experienced pandemic-related changes in screening and colposcopy):– Similar to the survey, the included qualitative interview items are part of a larger interview guide, of which details are lacking both on the overall and included interview items.

We have now added more details pertaining to qualitative interview guide items on pg. 6 of the manuscript.

“The final interview guide included in depth exploration of cervical cancer screening and management items explored in the quantitative survey. We more deeply explored screening practices (barriers and facilitators to screening for each clinician’s patient panel), adherence to 2019 ASCCP guidelines (how clinicians assess if patients are due for screening, type of screening test used, screening interval used and reasoning) barriers to adoption of ASCCP guidelines, HPV self-sampling (benefits and concerns), and the impact of the pandemic on screening and management practices. Additionally, there was a sub-set of questions for colposcopists (on colposcopy training, LEEP self-performance versus referral, biopsy location). This manuscript focuses on qualitative findings relevant to the COVID-19 pandemic and its impact on screening and abnormal results follow-up (pause and resumption of screening or follow-ups during pandemic, catching up on missed screenings).”

– How much overlap was there in the quantitative and qualitative components of the study? For example, between one of the three themes identified, the reductions in screening, and the survey responses.

When designing the qualitative interview guide, we intentionally selected items that expanded upon and provided in depth perspectives on quantitative areas of interest. We now describe this in text (see pg. 6) and have expanded on the explanation of both the quantitative survey items and qualitative interview guide as described above.

Reviewer #2 (Recommendations for the authors):Overall, I commend you highly for this important work. I am very impressed with the magnitude of this survey and the amount of demographic data acquired.I would recommend nuancing the finding of Black and Asian providers as having statistically significant reductions in cervical cancer screening since the confidence intervals cross one in both instances for these. OR I would also add in some discussion about the inherent weaknesses in a survey study as well as potential biases which may exist. I would also take out the comment about regional differences in the way the pandemic was handled in different geographic regions in the US unless you have a statistic to support this statement. I understand your passion, but this statement shows some biases by the authors which may be not well received in certain areas and take away from the importance of your work.Thank you for this well-done survey study. Your diligence and attention to detail in the statistical analysis are noteworthy. This is truly valuable public health information and a significant contribution to the field of cancer screening.

Thank you for your very kind review. In terms of the associations of race with perceived reduced screenings (and gender with reduced colposcopies), we have tempered the language in the discussion for findings where confidence intervals cross one and/or where p-values are between.05 and.10 (for race in the screening model, see pgs. 11 and 12; for gender in the colposcopy mode, see pg. 12) and discuss them with language indicating “marginal” associations. We have also added a sentence to the discussion focused on the potential bias of self-report surveys in relation to clinician practice outcomes:

“Self-report surveys have inherent biases and may not be actual representations of screening and colposcopy practices that could be ascertained via medical record or claims databases.”

Finally, we have removed the phrasing related to political handling of the pandemic in different regions as suggested and now focus this statement on the missing region data and its repercussions in our ability to detect associations:

“Next, 9% of the full sample did not respond to geographic location items, thus this gap in data may have limited our ability to detect differences surrounding region- and pandemic-related differences in screening and colposcopy.”

Reviewer #3 (Recommendations for the authors):Because the study was conducted in 2021, some questions/points should be addressed to increase the overall impact of the work:1) Highlight the context of the study: what was the situation in terms of the pandemic in the US when the study was conducted? This should be put in front in the introduction and results should be discussed regarding this context;

Thank you for the suggestion. We now include information about the context of the pandemic in the US in 2021 in the Method section:

“Data were collected between March-August 2021 (surveys) and June-December 2021 (interviews). For context, the COVID-19 vaccine became available to healthcare providers in the US in early 2021. The US general public had widespread access to vaccination beginning in the summer of 2021. By the fall of 2021, the pandemic appeared to be less acute in the US, with healthcare organizations attempting to resume normal operations through the end of the year. Masking, social distancing and reduced capacities indoors, and enhanced cleaning procedures were public health practices in place with varying levels of intensity across the US at this time.”

This information is now included on pg. 4 under Methods, Participant recruitment.

2) Results should be discussed in the light of what is already known/foreseen in terms of covid impact on cervical cancer screening and on cervical cancer;

Throughout the discussion (beginning on pg. 11) we cite the context of the COVID-19 pandemic in relation to cervical cancer screening, LEEP, and colposcopy practices. We have added a section on pg. 13 that discusses what is already known in terms of the impact of the pandemic on cancer screening overall, and cervical cancer screening.

*“*It is well known that cancer screenings decreased dramatically at the height of the pandemic, ^6-9^ with cervical cancer screening rates dropping in 2020 compared with previous years.^10,11^ As the pandemic progressed, cancer screening rates started to rebound,^6,14^ but our findings highlight challenges that still exist for cervical cancer screening and colposcopy.”

3) It would be interesting to discuss these results i.e. obtained from a high-income country, compared to what was reported from low- and medium-income countries e.g. in terms of disruption of services, active follow-up of patients (https://doi.org/10.1002/ijc.33500);

3) It would be interesting to discuss these results i.e. obtained from a high-income country, compared to what was reported from low- and medium-income countries e.g. in terms of disruption of services, active follow-up of patients (https://doi.org/10.1002/ijc.33500);

We now cite the similarities in the challenges experienced in cancer screening during the pandemic between what we found in this US-based study and low- and middle-income countries in the discussion on pg. 13.

4) Would results gained from this study be beneficial to improve cervical cancer screening or/and to health system responses to the pandemic?

Findings highlight challenges with screening and follow-up, but also serve as a metric for potential increases in cervical cancer incidence that may result from lags in return to screening and colposcopy. In particular, our findings highlight intervention points e.g., family and internal medicine physicians; community health/safety net settings of care. These implications are described in the discussion.

5) Results of the survey suggest that inequality of cervical cancer screening access might have increased due to covid pandemic; do the authors have any comment on it?

In terms of inequitable screening rates, we note potential disparities in the discussion on pg. 11:

“Concerningly, this may indicate worsening disparities in cancer prevention care in settings that serve patients with the highest cervical cancer rates: lower resourced and historically marginalized communities.^31^ These findings are supported by recent literature indicating that federally qualified health center settings suffered staffing losses and other challenges during the pandemic which led to reductions in cancer screenings due to postponement of preventive care.^32^”

The discussion should be revised to address the points listed above. I suggest also highlighting the pandemic context in the US when the study was conducted.

We have addressed the points above as suggested. We also highlight the context of the pandemic when data was collected on as noted in our first response to the reviewer. See pg. 4 (Method, Participant recruitment).